# Goose Meat as a Source of Dietary Manganese—A Systematic Review

**DOI:** 10.3390/ani13050840

**Published:** 2023-02-25

**Authors:** Zuzanna Goluch, Gabriela Haraf

**Affiliations:** Department of Food Technology and Nutrition, Faculty of Production Engineering, Wrocław University of Economics & Business, 53-345 Wrocław, Poland

**Keywords:** goose, meat, manganese, thermal treatment, adequate intake, reference values-requirements, PRISMA

## Abstract

**Simple Summary:**

Manganese is a trace element with many critical physiological functions, which should be supplied to animals and humans through diet. Since goose meat is eaten in many countries worldwide, this study aimed to systematically review the content of this element in goose meat and its relation to the recommended intake at the level of adequate intake (AI) and nutrient reference values-requirements (NRV-R). Already 100 g of goose meat can cover the daily AI per Mn for a wide range of adults, depending on the thermal treatment used. Placing information on the content of Mn in goose meat and the percentage of NRV-R on the packaging may be valuable information for the consumer in terms of making food choices to diversify the diet. Consumption of goose meat containing manganese may be justified in people struggling with mental disorders (depression, anxiety disorders), lipid (hypercholesterolemia), and carbohydrate metabolism (reduced glucose tolerance), in whom reduced concentration of this element has been confirmed in blood.

**Abstract:**

Manganese is a trace element with essential physiological functions that should be supplied to animals and humans through diet. Goose meat is prevalent in many regions of the world. Therefore, the aim of the study was a systematic review (PRISMA statement, 1980–2022) of the content of Mn in raw and cooked goose meat and their relation to the recommended intake at the level of adequate intake (AI) and the nutrient reference values-requirements (NRV-R). The literature analysis shows that the content of Mn in goose meat depends on the breed, type of muscles, the presence of skin, and the cooking method used. AI level recommendations for Mn intake range from 0.003 to 5.50 mg/day, depending on the country, age, and gender. Consumption by adults (regardless of sex) of 100 g of domestic or wild goose meat covers the daily AI per Mn in various percentages, depending on the type of muscles (more Mn in leg muscles), presence of skin (more Mn in skinless muscles), and thermal treatment (pan fried with oil, grilled, and cooked meat contains more). Placing information on the Mn content in goose meat and the percentage of NRV-R on the packaging may be valuable information for the consumer in making food choices to diversify the diet. There are few studies on the content of Mn in goose meat. Therefore, it is reasonable to conduct research in this area.

## 1. Introduction

Manganese (Mn) is one of the essential minerals in animal and human nutrition. It is classified as a micronutrient or trace element. Manganese (Mn) is an essential dietary mineral for mammals and is a component of metalloenzymes such as arginase, glutamine synthetase, and pyruvate carboxylase [1]. Because it activates manganese superoxide dismutase (MnSOD), manganese is necessary for normal antioxidant defenses. In addition, it is an activator for many hydrolases, kinases, decarboxylases, and transferases. Manganese is involved in amino acid-, lipid- and carbohydrate metabolism, and proteoglycan synthesis in bone formation. Its role in regulating and transforming thyroid hormones is indicated [2]. It is necessary for utilizing biotin, vitamin B1, and vitamin C. Metabolic association between manganese and choline affects fat metabolism in the liver [3]. Manganese competes directly with cobalt (Co) and iron (Fe) for binding sites in the digestive tract. Therefore, an excess of Co or Fe may result in lower absorption of Mn and its potential deficiency. The amount of manganese absorbed is inversely related to the concentration of manganese in the diet. The human body content of manganese is estimated to be 10–20 mg. The concentration is relatively high in bone and organs rich in mitochondria, such as the liver, pancreas, and kidney, and concentrations are low in muscle and plasma [4]. This regulation seems to be part of the adaptive changes to the amount of dietary manganese intake, which allow the maintenance of manganese homeostasis over a wide range of intakes.

Humans obtain manganese from the air, water, and food. Plant sources have much higher manganese concentrations than animal sources. Whole grains (wheat germ, oats, and bran), rice, and nuts (hazelnuts, almonds, and pecans) contain the highest amounts of manganese. Chocolate, tea, mussels, clams, legumes, fruit, leafy vegetables (spinach), seeds (flax, sesame, pumpkin, sunflower, and pine nuts), and spices (chili powder, cloves, and saffron) are also rich in manganese [5,6].

Manganese is mainly absorbed as Mn(II), and absorption is reported to be below 10% of ingested manganese. Manganese uptake in the intestine is mediated by high-affinity metal transporters, such as divalent metal transporter-1 (DMT1, also called DCT1), which is also involved in transporting other metals [7]. Absorbed manganese is transported to the liver and distributed to other tissues bound to transferrin, α2-macroglobulin, and albumin. The main route of elimination of manganese from the body is via bile to the small intestine, whereas very little is excreted in the urine. The half-life in men varies from 13 to 37 days, and the half-life in women is longer, but sizeable inter-individual variation exists [8]. Low dietary manganese intake results in increased manganese absorption relative to intake [9]. Vegetarians often have diets richer in manganese than those who select omnivorous diets because they consume more fiber and phytates. Dermal changes and hypercholesterolemia are possible signs of manganese deficiency, as well as diffuse bone demineralization and poor growth in children [10].

Geese are waterfowl consumed in some regions of the world, especially Asia, some countries of Europe, and the USA [11]. The quality of meat obtained from them is a complex combination of characteristics such as appearance, color, texture, functionality (e.g., cooking loss), taste, and nutritional value, including mineral content [12]. Goose meat can be a valuable source of minerals in the human diet [13]. However, their amount of meat depends on many factors: poultry species, age, sex of birds, rearing system and region, time of fattening, content in the feed, drinking water, premixes, and even medicines.

Deficiency of Mn in young birds causes perosis (swelling and deformation of the tibia-metatarsal joints, “slipping” of the Achilles tendon, and lameness). Manganese in plant materials is available in 50–60%, but its bioavailability in poultry feed mixtures is reduced in excess calcium and phosphorus. Because its content and bioavailability in plant feed fluctuate significantly, it is often added to premixes in the form of inorganic trace minerals (ITM), for example, oxides, carbonates, chlorides, and sulphates. Manganese has a low potential for toxicity due to its poor intestinal absorption and efficient biliary elimination. Still, it can interact with several other dietary nutrients, such as Zn and Fe, by competing with Fe for intestinal absorption sites or reducing Fe and Zn tissue concentrations [3]. Historically, manganese sulfate has been the most common source of manganese in animal supplements. Manganese sulphate monohydrate is generally used as the standard reference for assessing Mn bioavailability. Still, the actual absorption of the Mn in this compound is 2% to 8%, depending on the species and the diet to which Mn is added. Manganese can be added to feed in the form of organic trace minerals (OTM), for example, metal proteinate and metal amino acid chelate, which are more expensive than ITM. Relative bioavailability values for manganese in poultry can range from 29% for manganese dioxide MnO (ITM) to 174% for manganese methionine Mn-Met (OTM) compared to the sulphate standard [3]. However, an excessive supply of this ingredient in the feed reduces the absorption of iron, magnesium, and phosphorus. The minimum nutritional requirements of geese is 60 mg/kg in a grower and 40 mg/kg in a breeder.

The manganese content in goose feed, its chemical form, and bioavailability may be reflected in its range in goose meat. In the literature, there are few published research results on the content of Mn in goose muscles in terms of its consumption and, more often, in biomonitoring as a bioindicator of environmental pollution [14,15]. In addition, in the literature, the content of minerals in poultry meat is often given without dividing it into breast and leg meat, which as culinary portions, are most often consumed by consumers. However, muscles differ in their histological structure and the nature of metabolic changes, which may affect the content of minerals, including manganese [16]. Similarly, considering the sex of poultry, males, and females differ in the growth rate, which affects the amount of feed and manganese intake, and thus its use in the body, as well as excretion. Despite these facts, research conducted in the pectoral muscles of the Egyptian goose by Geldenhuys et al. [17] shows that the manganese content in males and females did not differ significantly, amounting to an average of 0.06 mg/100 g dry basis. In addition, as part of preventing cardiovascular disease or weight reduction, consumers are encouraged to remove the skin and subcutaneous fat from carcass elements before cooking. The skin is a source of fat, cholesterol, sulfur amino acids, collagen, elastin, fat-soluble vitamins, and minerals [18,19]. For the consumer, it may be essential to know the manganese content in the elements of a goose carcass with or without skin.

This systematic review aims to (1) analyze the original animal studies from 1980–2022 on the manganese content of raw and culinary processed goose, (2) present chosen recommendations for adequate intakes (AI) for manganese (mg/day) depending on age and sex; (3) compare manganese in goose meat content with adequate intake and nutrient reference values-requirements for humans.

## 2. Methods

The review was prepared following PRISMA guidelines to ensure the transparency of the research conducted [20]. An electronic-based search was performed in the scientific libraries Scopus, Web of Science, and ScienceDirect [21,22,23]. Searches comprised a combination of MeSH terms and keywords, applying quotes and field tags with BOOLEAN operators. For all databases, four primally exclusion steps were determined: 1. keywords (“goose AND meat”) AND (manganese or mineral OR Mn), 2. years (1980–2022), 3. language (English), 4. publication type (article).

The AGRO database was used as an additional source [24]. AGRO bibliographic database has been created by the members of the main library of the University of Life Sciences in Poznań since 1993. The database includes bibliographic descriptions of articles of 1010 Polish titles published in Polish and English.

The screening was made in the areas: title, abstract, keywords (Scopus), and topic (Web of Science, Science Direct). The search and selection process was performed by two reviewers working independently and in parallel. The results from each database were exported to files with the appropriate extension of CSV or excel, and summaries were created for each database containing publication information, including abstracts. The files are included in Appendix A. Initial searches returned 61 results, and 46 peer-reviewed papers were selected for consideration based on relevance after removing duplicates. Two independent researchers reviewed the title, abstracts, and papers that did not meet the criteria for inclusion in the review were excluded through discussion. Two independently working researchers analyzed the results obtained to avoid errors. Any inconsistencies were resolved through discussion. Exclusion criteria were as follows: reviews, research notes, book chapters, and no data on the content of manganese in goose meat in the article’s abstract and contents. Additional searches were performed on the FoodData Central U.S. website Department of Agriculture (USDA) [25]. A total of 11 original studies, 5 reports from the USDA database, and 2 reports from Tables of composition and nutritional value of food [26] were selected for review (Table 1). Figure 1 presents a diagram of the literature search and selection criteria. 

## 3. Manganese Content in Raw Goose Meat

The manganese content (mg/kg of tissue) in raw goose muscles is shown in Table 2. In Poland, a series of studies on the content of, for example, manganese, as an element necessary in animal organisms but also as a bioindicator of environmental pollution was conducted by Falandysz et al. [27,28,29,30,31,32]. In the studies carried out in the years 1978–1983, in the raw meat of geese for slaughter from northern Poland, 0.17 mg Mn/kg was found [27]; in 1984, 0.38 mg Mn/kg was found [28]; and in 1985, 0.26 mg Mn/kg was found [30]. In 1986, the average content of Mn in the raw muscles of slaughter geese from northern Poland was comparable to that found in earlier years (1983–1985) (1.9 mg/kg) [29], and in 1987 the mean value obtained was 0.27 mg Mn/kg [31]. In subsequent studies, in the meat of geese randomly selected from slaughterhouses during 1988–1991, Falandysz et al. [32] determined 0.25 mg Mn/kg. The content of manganese in the muscles of Polish geese, stated by the above-cited authors, was characteristic of this animal species and the type of tissue. Kunachowicz et al. [26] give 0.2 mg Mn/kg of tissue in the muscles of Polish geese. Chen et al. [34], examining goose meat purchased in markets and supermarkets in Taipei (Taiwan), found the average Mn content at 0.268 mg/kg of tissue. Considering the type of muscles, Oz and Celik [35] determined the manganese content in Turkish goose’s breast and leg muscles, which was 0.2 and 5.0 mg/kg tissue, respectively.

Considering the presence of skin, data from the U.S. Department of Agriculture) [25] show that raw goose with skin has less of this mineral than meat without skin (0.20 vs. 0.24 mg/kg tissue, respectively). Similarly, Goluch et al. [36] analyzed the Mn content in raw breast muscles with and without skin from White Kołuda^®^. The mean Mn content in breast muscles was 1.6 mg/kg dry mass (DM) and did not differ significantly between with skin and without skin (1.5 vs. 1.7 mg/kg DM).

Game geese are also a source of energy and nutrients for the population in many parts of the world. For example, in South Africa hunted (during spring and autumn hunting) Egyptian geese (*Alopochen aegyptiacus*) are eaten. In the Eastern James Bay Cree of Quebec in Canada, goose (*Branta canadensis*) is traditionally eaten by local rural communities. A study by Geldenhuys et al. [24] in the pectoral muscles of the Egyptian goose showed that the Mn content did not differ significantly in terms of sex and season (winter vs. summer) and was, on average, 0.6 mg/kg dry basis. Canada’s goose raw breast muscles without skin contained 0.50 mg Mn/kg of tissue [25].

## 4. Manganese Content in Goose Meat after Thermal Treatment

Because people rarely eat raw meat, the mineral content after it has been subjected to various thermal treatments is important. Cooking meat is essential to achieve a tasty and safe product. Different ways of processing meat strengthen its taste and delicacy and improve its hygienic quality by inactivating pathogenic microorganisms. During the heat treatment, cooking losses due to mass transfer depend on not only the cooking conditions, such as cooking method, cooking surface, cooking temperature, and time but also the meat properties, such as water content, fat content, protein content, pH value of the raw meat, and the meat portion size [37]. The internal temperature endpoint, during boiling, significantly affects mineral content [38]. Losses of minerals during the thermal treatment of meat depend on the form in which they occur. Mineral components, which can be found as soluble dissociated salts (part of sodium, small amounts of phosphorus, calcium, and potassium), go to the leakage. Components, such as iron, that combine with proteins remain in the meat [39]. The thermal treatment can lead to the loss of a part of the mineral matter, thereby reducing the product’s nutritional value. The most significant mineral matter reduction is generated when the meat is thermally treated in the aquatic environment [40].

The manganese content in goose meat subjected to thermal processing is presented in Table 3. USDA data analysis [25] shows the identical Mn content in raw and cooked skinless carcasses (0.24 mg/kg tissue), whereas in cooked carcasses with skin higher than in raw carcasses with skin (0.24 vs. 0.23 mg/kg tissue). The research conducted by Kunachowicz et al. [26] showed a higher content of Mn in cooked and roasted geese carcasses with skin than in raw carcasses (0.30 vs. 0.20 mg/kg tissue).

Oz and Celik [35] researched the breast and leg muscles of Turkish geese slaughtered at 24 weeks. Their estimates of manganese content were made both in raw meat (Table 2) as well as that submitted to boiling (<100 °C), grilling (180 °C), pan-frying without fat or oil (180 °C), pan-frying with oil (180 °C), deep-fat frying (180 °C), oven roasting (200 °C), and automatic microwave cooking. These various thermal treatments were applied for 5 to 35 min, depending on the cooking method and type of muscle. The content of Mn in the breast and leg muscles of geese was changed under the influence of various types of thermal processing; however, these changes were not statistically significant.

Goluch et al. [36] studied the effect of various methods of heat treatment (water bath cooking WBC, oven convection roasting OCR, grilling G, pan frying PF) on manganese content in White Kołuda^®^ goose breast muscles with and without skin. They found significantly (*p* ≤ 0.05) the highest Mn content in skinless WBC muscles compared to PF muscles (2.7 vs. 1.1 mg/kg DM). However, in muscles with skin, significantly (*p* ≤ 0.01), the highest content of Mn was found in grilled muscles (3.8 mg/kg DM), compared to raw muscles and other methods of heat treatment (raw 1.5, WBC 1.4, ORC 1.1, PF 1.3 mg/kg DM). In terms of Mn, the interaction between the type of breast muscle (with or without skin) and the heat processing method was statistically significant (*p* ≤ 0.001). However, the authors did not note significant differences in this ingredient retention, regardless of the muscle type (with or without skin) and the applied heat treatment.

Considering the wild goose, Geldenhuys et al. [33] found 1 mg Mn/kg of tissue in the breast muscles of Egyptian geese cooked in a preheated oven (160 °C). According to Sorbal et al. [39], boiling and frying lower the mineral content of meat, whereas baking, grilling, and microwaving increases it. Grilling and boiling pork and beef decrease the contents of Na, K, P, Ca, and Mg while increasing the contents of Fe and Zn [37]. Deep-frying, pan-frying, oven-cooking, and microwaving decrease the mineral content of cooked beefsteak, and microwaving causes the highest loss [41]. Purchas et al. [42] compared the mineral content in uncooked and cooked lean beef and reported a decrease in the contents of Na and K and an increase in Ca, Cu, Fe, Mn, and Zn in cooked meat in comparison with raw meat. These results indicate that divalent minerals are better retained during cooking than Na and K. The lower loss of divalent minerals during cooking is due to their greater association with protein. For manganese, Goluch et al. [36] observed no significant differences in the retention of this component, regardless of the type of goose muscle (skinless or with skin) and the applied heat treatment.

## 5. Recommendation for Manganese Intake and Its Consumption in Different Countries

Balance studies have suggested that an intake of 0.74 mg/d should be sufficient to replace daily manganese losses [43]. Intakes over 1 mg/d generally result in a positive manganese balance [10]. According to the data (Table 4) of the American Institute of Medicine (IOM, now National Academy of Medicine), the recommended intake of manganese for adults, at the level of adequate intake (AI), is 1.8 mg/day for women and 2.3 mg/day for men [44]. 

In 1993, the EU Scientific Committee for Food suggested 1–10 mg/d to be an acceptable intake of manganese [45]. The World Health Organization, the Food and Agriculture Organization of the United Nations (WHO/FAO), and the Nordic countries (Nordic Council of Ministers—NCM) have not established recommended intakes of manganese [46].

The European Food Safety Authority (EFSA) concluded that data is insufficient for deriving average requirements (ARs) or population reference intakes (PRIs) for manganese. EFSA proposed an adequate intake (AI) at 3 mg/day for adults based on observed mean intake from a mixed diet in the EU as stated in dietary reference values (DRVs) for manganese. An AI of 3 mg/day is proposed for adults, including pregnant and lactating women. For infants aged 7 to 11 months, an AI of 0.02–0.5 mg/day is suggested, reflecting the wide range of manganese intakes that appear to be adequate for this age group. The main contributor to dietary manganese intake is cereals (57%), followed by fruit, vegetables, nuts, and coffee/tea [9]. The societies for nutrition in Germany, Austria, and Switzerland recommend a manganese intake of 2.0–5.0 mg per day for adults and children above the age of 10 years [47]. Higher AI values have been determined in Australia and New Zealand: 5 mg/day for women and 5.5 mg/day for men [48]. Manganese standards in Poland have been adopted according to the IOM at the AI level [49].

The Scientific Committee on Food (SCF) could not set any observed adverse effect level (NOAEL) because no relevant dose-response animal studies were found. Consequently, SCF did not assign a tolerable upper intake level (UL) for manganese.

**Table 4 animals-13-00840-t004:** Adequate Intakes (AI) for manganese (mg/day) concerning the chosen recommendations.

Age, Both Sexes	IOM (2001) NIZP-PZH (2020) [44,49]	EFSA (2013) [9]		DACH (2021) [47]		NHMRC, AGDHA NZMH (2006) [46,48]
Infants						
0–6 months	0.003			4–12 months	0.6–1.0	0.003
7–12 months	0.6	7–11 months	0.02–0.5			0.600
Children and adolescents						
1–3 years	1.2	1–3 years	0.5	1–3 years	1.0–1.5	2.0
4–8 years	1.5	4–6 years	1	4–6 years	1.5–2.0	2.5
9–13 years	1.9 ♂ 1.6 ♀	7–10 years	1.5	7–9 years	2.0–3.0	3.0 ♂ 2.5 ♀
14–18 years	2.2 ♂ 1.6 ♀	11–14 years	2.0	≥10 years	2.0–5.0	3.5 ♂ 3.0 ♀
		15–17 years	3.0			
Adults						
>18 years	2.3 ♂ 1.8 ♀	≥18 years	3.0			5.5 ♂ 5.0 ♀
Pregnant all ages	2.0		3.0			5.0
Lactation all ages	2.6		3.0			5.0

AI—adequate intake; IOM—Institute of Medicine (now National Academy of Medicine); NIPH-NIH—National Institute of Public Health-National Institute of Hygiene; EFSA—European Food Safety Authority; NHMRC—National Health and Medical Research Council; AGDHA—Australian Government Department of Health and Ageing; NZMH—New Zealand Ministry of Health; DACH—Nutrition Societies in Germany and Austria and Switzerland.

In some countries, the intake of manganese is not often estimated. In the EU, adults’ estimated mean manganese intakes range from 2 to 6 mg/day, with most values around 3 mg/day. Estimated mean manganese intake ranges from 1.5 to 3.5 mg/day in children and 2 to 6 mg/day in adolescents [9]. The Polish population’s estimated intake of manganese was 4.7 mg/day. The average content of manganese in women’s diets was 4.10 mg/day, whereas in men’s diets, it was 5.45 mg/day [50]. 

In contrast, the Spanish population’s estimated daily manganese intake was 2.372 mg/day. The population of the island of El-Hierro presented the highest intake (2.717 mg/day), and the one in Fuerteventura (1.986 mg/day) showed the lowest intake [51]. Results from the Swedish Market Basket study, 2015, indicate an average daily manganese intake of 4.2 mg per person and day and was well above estimated adequate intakes (AI) of 1.8 mg per day for women and 2.3 mg per day according to the Institute of Medicine U.S. [44]. The average daily per capita intake of manganese from meat was 0.07 mg [52]. Calculations based on data from Denmark, 2013 and 2015, evaluate the mean dietary intake of manganese to 3.9 mg/day for adults and up to 6.9 mg/day in the higher intake groups. The manganese intake of Finnish children 3–18 years of age was 3–7 mg/d calculated from food consumption and composition [53]. In a study conducted among the communities of Northern Italy, the average intake of manganese was 2.34 (1.46–3.52) mg/day [54]. A pilot study of 20,000 people in Germany shows that the average intake of manganese in the general German population aged 14–80 is about 2.8 mg per day (for a 70 kg person) and is in the 2–5 mg per person per day as recommended by the German, Austrian and Swiss nutrition societies [55].

## 6. Coverage of the Adequate Daily Intake of Manganese (AI) and Nutrient Reference Values-Requirements by Goose Meat in Adults

For the consumer purchasing food products, the information on the label, which relates to energy and nutritional value, is essential. This information should also include the reference value of the daily intake (NRV). These recommendations are based on the best available scientific knowledge of the daily amount of energy or nutrient needed for good health [56]. The nutrient reference values (NRVs) for manganese, 2 mg/day for adults, is listed in Annex XIII to Regulation (EU) No 1169/2011 of The European Parliament and of The Council of 25 October 2011 on the provision of food information to consumer [57]. According to this regulation, the nutritional value of products, including mineral content, is given per 100 g or 100 mL. The Codex Committee on Nutrition and Foods for Special Dietary Uses has determined that NRV-R for manganese is 3 mg [58]. 

Table 5 presents calculations of the daily coverage of AI on Mn recommended by selected countries for adults, taking into account its content in raw and thermally treated meat. Consumption by adults (regardless of sex) of 100 g of raw Turkish goose leg muscles in the highest percentage (9.09–27.8%) will cover the daily AI per Mn, taking into account the presented recommendations of 1.8–5.5 mg/day. In the case of consumption of 100 g of raw breast muscle without skin, the highest percentage of AI implementation (3.09–9.44%) was calculated for meat from Polish White Kołuda^®^. However, considering wild goose meat, more AI will cover 100 g of raw muscle from Egyptian Geese (1.2–3.33%). In many countries, various thermal treatment of goose meat is used, which is associated with changes in the content of minerals and their retention [17,35,36]. According to our calculations, the largest percentage of the daily AI for adults will cover 100 g of Turkish goose leg meat fried in a pan with oil (32.5–81.5%) and Polish White Kołuda^®^ breast muscles grilled with skin (7.6–21.1%). Also, the consumption of 100 g of cooked wild Egyptian goose breast meat will cover a significant percentage (10.9–33.3%) of the daily AI. The higher Mn content in fried and pan-roasted meat is due to the specific nature of these treatments, which do not require water, allowing for greater water retention. It has already been shown that during frying, the meat surface temperature quickly reaches 115–120 °C or above 120 °C, and the protein forms a solid layer on the surface of the meat, thanks to which the loss of soluble substances such as inorganic salts is reduced [40].

Taking into account the marking of products with the NRV-R value (Table 5), raw goose meat without skin covers the daily demand of the consumer (regardless of gender) in 1.2–25% and with skin in 1.0–7.5% (in a higher percentage of leg meat). Manganese requirements for an adult are met to the greatest extent by pan-fried with oil (NRV-R = 81.5%) and grilled (NRV-R = 25%) Turkish goose leg muscles, as well as cooked muscles of domestic Turkish goose and wild Egyptian goose (NVR-R 25 to 30%). When analyzing the presence of skin, Polish White Kołuda^®^ goose grilled breast meat with skin covers 19% of the consumer’s NRV-R.

A deficiency of manganese in the diet can cause changes in the osteoarticular system. Research has now established that it is the most efficient divalent cation in activating the glycosyltransferase enzymes, essential in the chondroitin sulfate formation, the major polysaccharide of the cartilage synthesis of cartilage and bone collagen, as well as in bone mineralization. Women with osteoporosis have been shown to have lower serum Mn levels than women with normal bone mineral density [59].

A link between low dietary Mn and impaired insulin secretion and glucose metabolism has also been shown [60]. Mn is an essential component of MnSOD for reducing mitochondrial oxidative stress. Oxidative stress is a common risk factor for the development of metabolic diseases. Low manganese intake is correlated with metabolic syndrome (MS) and its components: abdominal obesity, hypertriacylglycerolaemia, and high blood pressure [1]. Manganese is essential for many vital processes, including nerve and brain development and cognitive function. In humans, manganese deficiency has also been linked with reduced levels of the neurotransmitter dopamine. However, excess Mn has neurotoxic effects [61].

Manganese is considered an essential and critical nutrient, and its deficiency or overabundance may cause depressive disorders. People with depression were observed to have lower urinary manganese concentrations or MnSOD activity compared to the control group [62]. Several studies have shown that manganese concentrations in patients with schizophrenia were lower than in control groups [63], but there are no consistent findings. That is why it seems important to balance the dietary content of this trace element.

## 7. Conclusions

The primary source of manganese in the human diet is tea and plant products. Diversifying the diet with goose meat is also worthwhile because, unlike plant food, it does not contain substances that diminish bioavailability, such as fiber or phytates. As a natural source of this element, goose meat does not increase exposure to overconsumption, as can happen when consuming dietary supplements. As little as 100 g of goose meat can cover the daily AI for Mn for a wide range of adults, depending on the heat treatment used.

Consumption of goose meat containing manganese may be justified in people suffering from mental disorders (depression, anxiety disorders), lipid (hypocholesterolemia), and/or carbohydrate (reduced glucose tolerance) disorders, in whom reduced concentration of this element has been confirmed in the blood. Placing information on the content of Mn in goose meat and the percentage of NRV-R on the packaging may be valuable information for the consumer in terms of making food choices to diversify the diet. Since there are few studies on the content of Mn in goose meat, it is reasonable to research the content of the impact of various thermal processing techniques on its concentration and retention.

## Figures and Tables

**Figure 1 animals-13-00840-f001:**
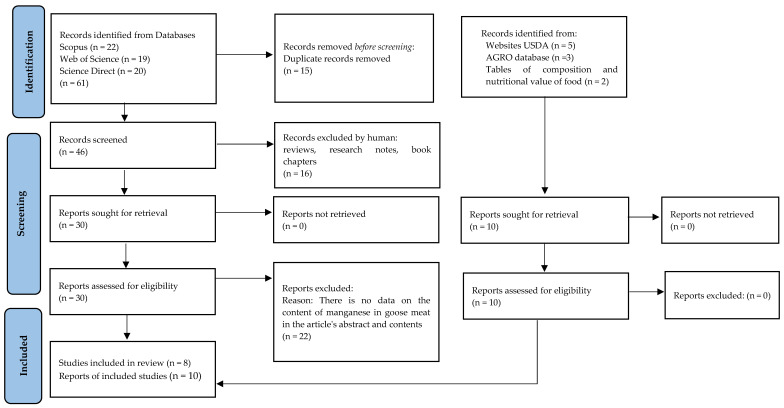
Identification, screening, and exclusion criteria of research included in the review.

**Table 1 animals-13-00840-t001:** Description of studies included in the systematic review.

First Author	Year	Source	References
Falandysz et al.	1986	Agro	[27]
Falandysz et al.	1987	Agro	[28]
Falandysz et al.	1989	Agro	[29]
Falandysz et al.	1989	Scopus	[30]
Falandysz J.	1991	Scopus, WoS	[31]
Falandysz et al.	1994	Scopus, WoS	[32]
Geldenhuys et al.	2013	Scopus, WoS	[33]
Geldenhuys et al.	2015	Scopus, WoS, Science Direct	[17]
Chen et al.	2013	Scopus, WoS	[34]
Oz and Celik	2015	Scopus	[35]
Goluch et al.	2021	Scopus, WoS, Science Direct	[36]
Kunachowicz et al.	2020	Tables of composition and nutritional value of food.	[26]
USDA Food Data Central	2022	Websites	[25]

**Table 2 animals-13-00840-t002:** Manganese content of goose meat raw (mg/kg tissue).

Meat	Goose	n	Age	Carcass	Breast	Leg/Thigh	Reference
Meat raw	*Anser Anser*	29	n.d.	0.17 ± 0.04 (0.12–0.24)	n.d.	n.d.	[27]
Meat raw	*Anser Anser*	18	n.d.	0.38 (0.14–2.41)	n.d.	n.d.	[28]
Meat raw	*Anser Anser*	18	n.d.	0.25 (0.17–0.31)	n.d.	n.d.	[30]
Meat raw	*Anser Anser*	39	n.d.	0.26 (0.14–0.48)	n.d.	n.d.	[29]
Meat raw	*Anser Anser*	32	4–7 months	0.27 (0.15–0.42)	n.d.	n.d.	[31]
Meat raw	*Anser Anser*	16	4–7 months	0.25 ± 0.06 (0.15–0.30)	n.d.	n.d.	[32]
Meat raw	*Anser Anser*	n.d.	n.d.	0.20	n.d.	n.d.	[26]
Meat raw	*Anser Anser*	10	n.d.	0.268 ± 0.073 (0.151–0.367)	n.d.	n.d.	[34]
Meat only, raw	*Anser Anser*	n.d.	n.d.	0.24	n.d.	n.d.	[25]
Meat and skin, raw	*Anser Anser*	n.d.	n.d.	0.20	n.d.	n.d.	[25]
Meat raw	Turkish goose	16	16 weeks	n.d.	0.2	5.0	[35]
Meat raw:	White Kołuda^®^	24	17 weeks	n.d		n.d	[36]
Without skin					1.7		
With skin					1.5		
Meat raw	Egyptian goose *Alopochen aegyptiacus*	36	n.d.	n.d.	0.6 ± 0.01 ♀♂ 0.6 ± 0.08 season winter/summer	n.d.	[17]
Meat raw, skinless	Canada goose *Branta canadensis*	6	n.d.	0.50	n.d.	n.d.	[25]

n.d.—no data.

**Table 3 animals-13-00840-t003:** Manganese content of goose meat after thermal treatment (mg/kg tissue).

Meat	Goose	n	Age	Carcass	Breast	Leg/Thigh	Reference
Meat only, cooked, roasted	*Anser Anser*	n.d.	n.d.	0.24	n.d.	n.d.	[25]
Meat and skin, cooked, roasted	*Anser Anser*	n.d.	n.d.	0.23	n.d.	n.d.	[25]
Meat and skin, cooked, roasted	*Anser Anser*	n.d.	n.d.	0.30	n.d.	n.d.	[26]
Meat:	Turkish goose	16	16 weeks	n.d.			[35]
Boiled					0.7 ± 0.2	8.0 ± 8.0	
Grilled					0.4 ± 0.1	5.0 ± 3.0	
Pan fried without fat or oil					4.5 ± 6.0	3.4 ± 3.1	
Pan with oil					0.5 ± 0.5	16.3 ± 18.1	
Deep-fat fried					0.6 ± 0.7	2.1 ± 0.3	
Oven cooked					0.3 ± 0.3	2.4 ± 0.9	
Microwave					0.1 ± 0.1	1.15 ± 2.0	
Meat:	White Kołuda^®^	48	17 weeks	n.d.		n.d.	[36]
Water bath cooking							
Without skin					2.7		
With skin					1.4		
Grilled							
Without skin					2.0		
With skin					3.8		
Oven convection Roasting							
Without skin					1.3		
With skin					1.1		
Pan fried							
Without skin					1.1		
With skin					1.3		
Meat breast, cooked	Egyptian goose *Alopochen aegyptiacus*	6	n.d.	n.d.	1.0	n.d.	[33]

n.d.—no data.

**Table 5 animals-13-00840-t005:** Percentage of the coverage of the AI for manganese of adults by the consumption of 100 g of goose meat, with regard chosen the recommendation, and nutrient reference values-requirements.

Meat	Goose	Manganese Content [mg/100 g]	Reference	IOM (2001) NIPH-NIH (2020) [44,49]		EFSA (2013) [9]	NHMRC, AGDHA NZMH (2006) [46,48]		DACH (2021) [47]	NRV-R [57]
				♀ 1.8 mg	♂ 2.3 mg	♀♂ 3.0 mg	♀ 5.0 mg	♂ 5.5 mg	♀♂ 2.0–5.0 mg	♀♂ 2.0 mg
Raw										
Meat raw, only	*Anser Anser*	0.024 carcass	[25]	1.33	1.04	0.8	0.48	0.44	1.2–0.48	1.2
Meat and skin, raw	*Anser Anser*	0.020 carcass	[25]	1.11	0.86	0.67	0.4	0.36	1.0–0.4	1.0
Meat raw	Turkish	0.020 breast 0.50 leg	[35]	1.11 27.8	0.86 21.7	0.86 16.7	0.4 10.0	0.36 9.09	1.0–0.4 10.0	1.0 25.0
Meat raw without skin	White Kołuda^®^	0.17 breast	[36]	9.44	7.39	5.67	3.4	3.09	8.5–3.4	8.5
Meat raw with skin	White Kołuda^®^	0.15 breast	[36]	8.33	6.52	5.0	3.0	2.73	7.5–3.0	7.5
Meat raw, skinless	Canada goose	0.05 carcass	[25]	2.78	2.17	1.67	1.0	0.91	2.5–1.0	2.5
Meat raw	Egyptian goose	0.06 ♂♀	[17]	3.33	2.60	2.0	1.2	10.9	3.0–1.2	3.0
Thermal treatment										
Meat only, cooked, roasted	*Answer Anser*	0.024 carcass	[25]	1.33	1.04	0.8	0.48	0.44	1.2–0.48	1.2
Meat and skin, cooked, roasted	*Anser Anser*	0.023 carcass	[25]	1.28	1.0	0.77	0.46	0.42	1.15–0.46	1.15
Meat:	Turkish goose	Breast/leg	[35]							
Boiled		0.07/0.08		3.89/4.44	3.04/3.48	2.33/2.67	1.4/1.6	1.27/1.27	3.5/4.4–1.4/1.6	3.5/4.0
Grilled		0.04/0.50		2.22/27.8	1.74/2.17	1.33/16.7	0.8/10.0	0.73/9.09	2.0/25.0–0.8/10.0	2.0/25.0
Pan fried without fat or oil		0.45/0.34		25.0/18.9	19.6/14.8	15.0/11.3	9.0/6.8	8.18/6.18	22.5/17.0–9.0/6.8	22.5/17.0
Pan with oil		0.05/1.63		2.78/90.6	2.17/70.9	1.67/54.3	1.0/32.5	0.91/29.6	2.5/81.5–1.0/32.5	2.5/81.5
Deep-fat fried		0.06/0.21		3.33/11.7	2.61/9.13	2.0/7.0	1.2/4.2	1.09/3.82	3.2/10.5–1.2/4.2	3.0/10.5
Oven cooked		0.03/0.24		1.67/13.3	1.30/10.4	1.0/8.0	0.6/4.8	0.55/4.36	1.5/12.0–0.6/4.8	1.5/12.0
Microwave		0.01/0.15		0.56/3.33	0.43/6.52	0.33/5.0	0.2/3.0	0.18/2.73	0.5/7.5–0.2/3.0	0.5/7.5
Meat:	White Kołuda^®^	breast	[36]							
Water bath cooking										
Without skin		0.27		15.0	11.7	9.0	5.4	4.91	13.5–5.4	13.5
With skin		0.14		7.8	6.09	4.67	2.8	2.55	7.0–2.8	7.0
Grilled										
Without skin		0.20		11.1	8.70	6.67	4.0	4.0	10.0–4.0	10.0
With skin		0.38		21.1	16.5	12.7	7.6	6.91	19.0–7.6	19.0
Oven convection Roasting										
Without skin		0.13		7.22	5.65	4.33	2.6	2.36	6.5–2.6	6.5
With skin		0.11		6.11	4.78	3.67	2.2	2.0	5.5–2.2	5.5
Pan fried										
Without skin		0.11		6.11	4.78	3.67	2.2	2.0	5.5–2.2	5.5
With skin		0.13		7.22	5.65	4.33	2.6	2.36	6.5–2.6	6.5
Meat breast, cooked	Egyptian goose	breast	[17]							
		0.60		33.3	26.1	20.0	12.0	10.9	30.0-12.0	30.0

AI—adequate intake; IOM—Institute of Medicine (now: National Academy of Medicine); NIPH-NIH—National Institute of Public Health-National Institute of Hygiene; EFSA—European Food Safety Authority; NHMRC—National Health and Medical Research Council; AGDHA—Australian Government Department of Health and Ageing; NZMH—New Zealand Ministry of Health; DACH—Nutrition Societies in Germany and Austria and Switzerland; NRV-R—nutrient reference values-requirements.

## Data Availability

Not applicable.

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
