# Peer review of "Goose Meat as a Source of Dietary Manganese—A Systematic Review"

_animals, 2023, doi:10.3390/ani13050840_

Round 1

Reviewer 1 Report

Comment for authors

The present study “Goose meat as a source of dietary manganese” is valuable for the reader and has practical significance. However, I would suggest to modify the title having the term meta-analysis/review of literature. There are some other points that should be considered before it publications.

·         Lines 40-42: please add reference for this statement. Same for line 50-53

·         I would suggest to write some other sources of Mn from foods in introduction section.

·         Line 107-108: it would be great if authors mention which sex has more Mn content.

·         Objectives should be written more in details.

·         Section 3: authors should mention why there is loss of Mn during thermal treatment of meat.

·         Authors should add more discussion as they indicated in abstract and conclusion regarding the disorders caused by the deficiency of Mn.

Reviewer 2 Report

There are minor corrections pointed out in attached file

Reviewer 3 Report

see attached document

Round 2

Reviewer 1 Report

no further concerns

Reviewer 2 Report

.